# REVISITING GRAPH CONTRASTIVE LEARNING THROUGH THE LENS OF CONTRASTIVE OVERFITTING

## ABSTRACT

Graph Contrastive Learning (GCL) has emerged as a powerful framework for unsupervised graph representation learning, typically optimized with contrastive objectives such as InfoNCE. Contrary to the common belief that lower contrastive loss implies better representations generated for downstream tasks, we observe little positive correlation between the contrastive objective and downstream performance. In fact, excessive optimization often leads to degraded performance–a clear symptom of overfitting. We attribute this phenomenon to the structure-agnostic nature of contrastive objective, which forces the encoder to discard essential structural information. Through extensive empirical and theoretical studies, we verify that the overfitted embeddings, which scarcely capture graph structural information, substantially impair generalization when applied to downstream classifiers. To address this issue, we propose a structure-preserving regularization (SPR) framework that can be seamlessly integrated as a plug-and-play module to enhance existing GCL methods. Comprehensive experiments across multiple datasets and baselines demonstrate that our approach effectively mitigates the overfitting problem.

## 1 INTRODUCTION

Graph Contrastive Learning (GCL), which extends contrastive learning techniques to graph-structured data, has emerged as a promising paradigm for graph representation learning, particularly due to its ability to learn without manually annotated labels (Liu et al., 2022; Ju et al., 2024). The primary objective of GCL is to train an encoder–typically a Graph Convolutional Network (GCN) (Kipf & Welling, 2017)– to generate node embeddings that are both informative and discriminative for downstream tasks such as node classification. Among various approaches, optimization objectives to minimize the InfoNCE-based loss have become the mainstream.

Although previous works report favorable downstream classification performance gains by using InfoNCE-based optimization objectives (Zhu et al., 2020; 2021a;b), closer inspection highlights a fundamental misalignment between contrastive objective and downstream task. Ideally, a well-optimized GCL objective should lead to representations that yeild better downstream performance. However, empirical evidence suggests this correspondence is inconsistent: a better-converged contrastive objective does not necessarily yield better-performing representations and can, in fact, degrade performance. We refer to this misalignment phenomenon as *contrastive overfitting*.

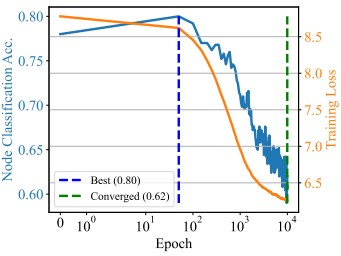

Figure 1: GCL loss and node classification accuracy over training epochs, using GRACE (Zhu et al., 2020) on PubMed (Yang et al., 2016) dataset.

As shown in Figure 1, while the contrastive loss decreases monotonically to convergence, downstream performance follows a clear rise-then-fall trajectory during GCL training–a hallmark of overfitting, which is consistent on other datasets and models as shown in Appendix B. Crucially, this issue is inevitable in the unsupervised setting: with labels unavailable for the training data, there is no validation set for early stopping or hyper-parameter tunning. As a result, the number of training iterations is typically determined by extensive trial-and-error experiments. We refer to this as *evaluation bias*, which leads prior work to overlook the misalignment.

This anomalous behavior in GCL raises two fundamental questions: *what causes this overfitting, and how can it be effectively mitigated?* In this paper, we ascribe such overfitting to the structure-agnostic nature of the GCL training objective. Specifically, optimizing such an objective overlooks the structural roles of nodes and ultimately compels the encoder to ignore the graph's structural information, which is essential for graph representation learning. It is worth noting that some existing studies, such as (Xia et al., 2022), tackle the problem of false negatives in GCL loss and suggest that fitting to these samples undermines performance. Which aligns to some extent with our claim. Nevertheless, their improvements are still drawn from biased evaluations and can not generalize to the overfitting scenario. Further discussions about related works can be found in Appendix A.

Motivated by the insight that contrastive objectives are inherently structure-agnostic and thus fail to encode essential graph structural information, we introduce a structure-preserving framework that takes node embeddings as input and ensures that the outputs align with both local context and global structural equivalence of nodes, guided by mutual inference and graph centrality measures. In addition, after the regularized training, we further employ a post-hoc structural augmentation technique that directly injects structural information into the learned node embeddings. Together, these strategies effectively preserve structural signals and mitigate contrastive overfitting.

We highlight **our contributions** as follows:

- We identify that a well-optimized encoder in GCL can actually produce node representations with poor downstream performance, revealing a previously overlooked issue of contrastive overfitting in GCL.

- We conducted extensive investigations and verified, both theoretically and empirically, that the structure-agnostic nature of the contrastive objective is the key factor underlying contrastive overfitting. Specifically, optimizing a structure-agnostic contrastive loss prevents the encoder from capturing graph structural information, which inherently carries label-discriminative signals, thereby leading to insufficient encoding of label information.

- To address this issue, we propose a regularization approach that explicitly ensures the graph structural information is encoded. Extensive experiments validate the effectiveness of our method across multiple datasets and GCL baselines.

## 2 PRELIMINARIES

**Notations.** We denote an undirected graph as $\mathcal{G} = (\mathcal{V}, \mathcal{E}, \mathbf{X}, \mathbf{Y})$, where $\mathcal{V} = \{i\}_{i=1}^{N}$ represents the set of $N$ nodes, $\mathcal{E} \subseteq \mathcal{V} \times \mathcal{V}$ denotes the set of edges, $\mathbf{X} \in \mathbb{R}^{N \times D}$ is the given node attribute (feature) matrix, where each row $\mathbf{x}_i \in \mathbb{R}^{D}$ corresponds to the feature vector of node $i$. $\mathbf{Y} \in \{0, 1, \dots, K\}^{N}$ denotes the labels of all nodes. Let $\mathbf{A} \in \{0, 1\}^{N \times N}$ be the adjacency matrix, where $\mathbf{A}_{i,j} = 1$ if $(i, j) \in \mathcal{E}$, and $\mathbf{A}_{i,j} = 0$ otherwise. The neighborhood of node $i$ is defined as the set of its adjacent nodes: $\mathcal{N}_i = \{j \mid (i, j) \in \mathcal{E}\}$. $\mathbf{Z} \in \mathbb{R}^{N \times d}$ is the node embedding matrix, where $d$ denotes the dimensionality of latent space.

**Unsupervised Graph Representation Learning.** Graph representation learning focuses on training a GNN encoder to generate informative node embeddings, which are subsequently passed to a downstream classifier for task prediction. Formally, the process can be represented as follow:

$$\hat{\mathbf{Y}} = g_\psi \left( f_\phi \left( \mathbf{A}, \mathbf{X} \right) \right),$$

where $f_\phi : \{0, 1\}^{N \times N} \times \mathbb{R}^{N \times D} \mapsto \mathbb{R}^{N \times d}$ is the GNN encoder, $g_\psi : \mathbb{R}^{N \times d} \mapsto \{0, 1, \dots, K\}^{N}$ is the downstream classifier, $\hat{\mathbf{Y}}$ is the predicted node labels. Unsupervised graph representation learning (e.g., graph contrastive learning) typically follows a two-stage optimization procedure: first, the encoder $f_\phi$ is optimized by minimizing an unsupervised loss; then, fixing the optimal encoder $f_{\phi^*}$, the downstream classifier $g_\psi$ is trained by empirical risk minimization. Notably, node labels are unavailable during the unsupervised training of the encoder.

**Graph Contrastive Learning.** GCL aims to learn high-quality node embeddings by contrasting different augmented views of a graph. The framework typically involves three steps: graph augmentation, encoding, and contrasting. First, multiple graph views are generated using random augmentation

techniques. Then, these views are passed through a shared GNN encoder to produce node embeddings. Finally, the embeddings are used to compute a contrastive loss, which is minimized to update the encoder parameters. The training objective of GCL is to bring representations of positive pairs closer while pushing apart those of negative pairs, which can be achieved by optimizing the following widely-adopted InfoNCE-based contrastive loss (Oord et al., 2018; Zhu et al., 2020):

$$\mathcal{L}_{\text{con}} = \frac{1}{2N} \sum_{i \in \mathcal{V}} \left( \mathcal{L}(\mathbf{u}_i, \mathbf{v}_i) + \mathcal{L}(\mathbf{v}_i, \mathbf{u}_i) \right), \tag{1}$$

where $\mathbf{u}_i$ and $\mathbf{v}_i$ are embeddings of node $i$ in augmented views $\mathcal{G}_U$ and $\mathcal{G}_V$. $\mathcal{L}(\mathbf{u}_i, \mathbf{v}_i)$ is defined as

$$\mathcal{L}(\mathbf{u}_i, \mathbf{v}_i) = -\log \frac{e^{\theta(\mathbf{u}_i, \mathbf{v}_i)/\tau}}{e^{\theta(\mathbf{u}_i, \mathbf{v}_i)/\tau} + \sum_{j \in \mathcal{V}/i} e^{\theta(\mathbf{u}_i, \mathbf{v}_j)/\tau} + \sum_{j \in \mathcal{V}/i} e^{\theta(\mathbf{u}_i, \mathbf{u}_j)/\tau}}, \tag{2}$$

and $\mathcal{L}(\mathbf{v}_i, \mathbf{u}_i)$ is symmetric with respect to Equation (2). In Equation (2), $\theta$ is a similarity measure function, and $\tau$ is the temperature coefficient.

## 3 EMPIRICAL INVESTIGATION AND THEORETICAL ANALYSIS

### 3.1 MOTIVATING HYPOTHESIS

GCL aims to encode as much label-relevant information as possible into node embeddings, which typically arises from two sources: a node's intrinsic attributes and its structural role within the graph. As a non-Euclidean data with disordered and variable number of neighbors, a graph's topology inherently carries rich information about node relationships, community structures, and functional roles, making structural information essential for effective representation learning. However, conventional instance-level contrastive objectives, such as InfoNCE, primarily focus on aligning representations of the same node across different views and distinguishing them from others, without explicitly modeling structural dependencies. Consequently, during training, the model can achieve low contrastive loss by relying largely on node attributes, potentially overlooking structural cues. This results in embeddings that capture node-level similarity but underrepresent the graph's structural roles, which may limit their utility for downstream tasks.

### 3.2 EMPIRICAL OBSERVATIONS

In this section, we conduct detailed investigations of the contrastive overfitting in GCL. We test multiple existing GCL methods, and obtain several important yet counter-intuitive findings as follow:

**Observation 1: GCL methods with structure-agnostic objective always suffer from contrastive overfitting.** We evaluate four representative node-level GCL methods: GRACE (Zhu et al., 2020), ProGCL (Xia et al., 2022), CCA-SSG (Zhang et al., 2021), and DGI (Veličković et al., 2018). Specifically, GRACE and ProGCL adopt InfoNCE-based objectives, CCA-SSG employs a feature-level objective derived from canonical correlation analysis (CCA), and DGI maximizes the mutual information between local and global graph representations. The details are summarized in Table 1.

Table 1: Statistics of GCL methods.

| Method | contrastive object | contrastive strategy | structure-agnostic |
|---|---|---|---|
| GRACE | Equation (1) | InfoNCE-based | ✓ |
| ProGCL | Equation (1) with false negative weights | InfoNCE-based | ✓ |
| CCA-SSG | $\|\mathbf{U} - \mathbf{V}\|^2 - \lambda \left( \|\mathbf{U}^\top \mathbf{U} - \mathbf{I}\|^2 + \|\mathbf{V}^\top \mathbf{V} - \mathbf{I}\|^2 \right)$ | CCA-based | ✓ |
| DGI | $-\frac{1}{N+M} \sum_{i=1}^{N} \mathbb{E}_{\mathcal{G}}[\log(\mathcal{D}(\mathbf{h}_i, \mathbf{s})] + \sum_{i=1}^{M} \mathbb{E}_{\tilde{\mathcal{G}}}[\log(1 - \mathcal{D}(\tilde{\mathbf{h}}_j, \mathbf{s}))]$ | InfoMax-based | |

\* $\mathbf{U}$ and $\mathbf{V}$ are node embeddings of graph view $\mathcal{G}_U$ and $\mathcal{G}_V$; $\tilde{\mathcal{G}}$ is the corrupted graph of $\mathcal{G}$.

As shown in Figure 2, the node classification accuracy of GRACE, ProGCL, and CCA-SSG drops sharply when the encoder converges, whereas DGI maintains stable accuracy without exhibiting contrastive overfitting. We attribute this performance degradation to the structure-agnostic nature of

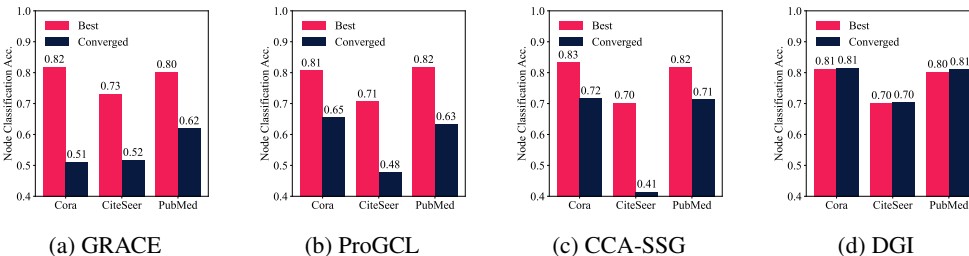

Figure 2: Performance degradation of GRACE, ProGCL, CCA-SSG, and DGI on different datasets.

their optimization objectives. In contrast, DGI contrasts local and global representations, compelling the discriminator to learn their matching relationships. This design naturally enforces the model to preserve information that differentiates augmented views, thereby enabling the encoding of structural information. Similar observations for other InfoNCE-based methods can be found in Appendix B.

**Observation 2: Graph structural information tends to be discarded during training.** Unlike contrastive learning methods designed for Euclidean data (Chen et al., 2020; Oord et al., 2018; Tschannen et al., 2020), GCL places a much greater reliance on the encoder's ability to capture and represent high-dimensional, non-Euclidean structures, particularly when dealing with non-attributed graphs, where all discriminative information is derived from the graph structure.

Let $\mathbf{H} \in \mathbb{R}^{N \times D}$ denotes the node representations output by a hidden layer in GNN encoder, $\mathbf{W}$ is a trainable matrix, $l$ is the layer index, the simplified expression of one layer of GNN and MLP are as follow:

$$\textbf{GNN:} \quad \mathbf{H}^{(l+1)} = \sigma\left(\mathbf{A}\mathbf{H}^{(l)}\mathbf{W}^{(l)}\right), \qquad \textbf{MLP:} \quad \mathbf{H}^{(l+1)} = \sigma\left(\mathbf{H}^{(l)}\mathbf{W}^{(l)}\right). \qquad (3)$$

It can be observed that, different from MLP, GNN is structure-aware, which stems from the left multiplication of adjacency matrix, known as message-passing mechanism (Gilmer et al., 2017).

To quantify the structural information encoded by GNN encoder, we employ the sensitivity of GNN to graph structural perturbations as a proxy metric. Specifically, we randomly drop edges of the original graph $\mathcal{G}$ to generate a corrupt graph $\tilde{\mathcal{G}}$, with the adjacency matrix $\tilde{\mathbf{A}}$. $\mathbf{Z} = f_\phi(\mathbf{A}, \mathbf{X})$ and $\tilde{\mathbf{Z}} = f_\phi(\tilde{\mathbf{A}}, \mathbf{X})$ are the embedding matrices. We define $\mathcal{C} = 1 - \frac{1}{N}\sum_{i=1}^N \frac{\mathbf{z}_i \tilde{\mathbf{z}}_i^\top}{\|\mathbf{z}_i\| \cdot \|\tilde{\mathbf{z}}_i\|}$ as a proxy metric to quantify how much structural information is captured, which calculates the average cosine similarity between the embeddings of the same node before and after graph corruption. For a structure-agnostic encoder such as MLP, node embeddings remain unchanged after corruption, yielding an averaged cosine similarity of 1, resulting in $\mathcal{C} = 0$. In general, if node embeddings change little after graph corruption, it indicates that the encoder captures less structural information, corresponding to $\mathcal{C}$ close to 0.

Based on this proxy metric, we track both the value of $\mathcal{C}$ and the loss throughout GCL training. As shown in Figure 3, the training loss is strongly positively correlated with the proxy graph information metric $\mathcal{C}$, indicating that the encoder becomes increasingly insensitive to changes in the graph structure during training, and gradually discards structural information.

**Observation 3: Structural dependency governs the degree of contrastive overfitting.** For different datasets, we assess their dependency on graph structural information by removing the structure-encoding capability, i.e., replacing the GCN encoder with an MLP, and then observing the their classification accuracy.

As shown in Table 2, we find that some datasets exhibit a strong dependency on graph structural information, where replacing the GCN encoder with an MLP significantly degrades downstream performance (e.g., Cora, CiteSeer, and PubMed). In contrast, some datasets can still maintain stable node classification performance even when using an MLP encoder (e.g., Am-Photo, Co-CS, and Wiki-CS). Moreover, the last row in Table 2 shows that, on the Am-Photo, Co-CS, and Wiki-CS datasets, the classifier trained directly on raw node attributes, also achieves comparable performance.

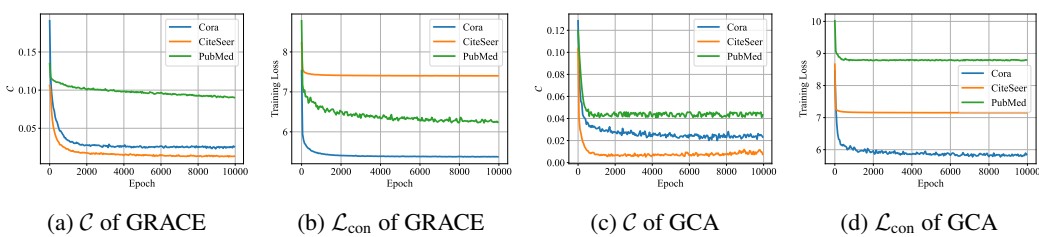

| (a) $\mathcal{C}$ of GRACE | (b) $\mathcal{L}_{\mathrm{con}}$ of GRACE | (c) $\mathcal{C}$ of GCA | (d) $\mathcal{L}_{\mathrm{con}}$ of GCA |

Figure 3: Graph structural information of GRACE during training.

Table 2: The performance comparison between using MLP and GCN as the GCL encoder.

| Method | Cora | CiteSeer | PubMed | Am-Photo | Co-Cs | Wiki-CS |
|---|---|---|---|---|---|---|
| GRACE | $81.30_{\pm 0.90}$ | $73.12_{\pm 0.30}$ | $80.12_{\pm 0.10}$ | $91.95_{\pm 0.02}$ | $92.25_{\pm 0.18}$ | $79.81_{\pm 0.04}$ |
| GRACE$_{\mathrm{MLP}}$ | $60.85_{\pm 0.05}\ 20.45\downarrow$ | $63.70_{\pm 1.00}\ 9.42\downarrow$ | $75.70_{\pm 0.10}\ 4.42\downarrow$ | $90.72_{\pm 0.02}\ 1.23\downarrow$ | $92.12_{\pm 0.04}\ 0.13\downarrow$ | $78.12_{\pm 0.10}\ 1.69\downarrow$ |
| Converged GRACE | $51.00_{\pm 0.00}\ 31.30\downarrow$ | $51.68_{\pm 0.41}\ 21.44\downarrow$ | $61.88_{\pm 0.86}\ 18.24\downarrow$ | $90.28_{\pm 0.15}\ 1.67\downarrow$ | $84.90_{\pm 0.08}\ 8.05\downarrow$ | $79.03_{\pm 0.05}\ 0.78\downarrow$ |
| Raw Features | $47.49_{\pm 0.13}\ 33.81\downarrow$ | $49.47_{\pm 0.63}\ 23.65\downarrow$ | $69.34_{\pm 0.74}\ 10.18\downarrow$ | $87.79_{\pm 0.89}\ 4.16\downarrow$ | $91.17_{\pm 0.30}\ 1.08\downarrow$ | $74.84_{\pm 0.42}\ 4.97\downarrow$ |

This suggests that label-relevant information in these datasets is largely derived from node attributes. Meanwhile, and more importantly, we can observe that, compared to structure-sensitive datasets, those are less reliant on structural information exhibit a lower degree of overfitting.

## 3.3 THEORETICAL INSIGHTS

To understand this phenomenon more fundamentally, we analyze the underlying conflict between the contrastive learning objective and the message-passing mechanism. Specifically, GNN captures structural information through message-passing, which inherently promotes local smoothness in node representations. For instance, a $k$-layer GNN aggregates information from $k$-hop neighborhoods. This process can be viewed as minimizing the graph Laplacian energy:

$$E(\mathbf{Z}) = \tfrac{1}{2} \sum_{(i,j)\in\mathcal{E}} \|\mathbf{z}_i - \mathbf{z}_j\|^2, \tag{4}$$

which encourages adjacent nodes to be close in the embedding space. In contrast, the InfoNCE loss promotes global separability: embeddings of different nodes are pushed apart regardless of their structural proximity.

In summary, message passing promotes local smoothness by encouraging structurally close nodes to have similar representations, while InfoNCE enforces global separability by pushing all node embeddings apart to maximize discrimination. These two objectives are fundamentally at odds. When the InfoNCE loss is overly optimized, the separation effect dominates, gradually diminishing the encoder's sensitivity to structural perturbations. This inherent conflict aligns with our empirical findings: as training progresses, the influence of message passing is progressively suppressed, leading to a reduced capacity of the encoder to capture and reflect structural variations.

## 4 THE PROPOSED METHOD

From the Bayesian perspective, we propose a Structure-Preserving Regularization (SPR) framework to mitigate contrastive overfitting, which introduces the structural prior. This prior constrains the encoder to capture structural information by aligning node embeddings with both local connectivity patterns and global structural roles. In particular, it keeps the embeddings of nodes with equivalent structure more similar, as well as embeds structural role cues that ensures properties such as node centrality to be inferred from the learned representations. In addition, we propose a simple yet effective parameter-free post-hoc embedding enhancement mechanism, which directly improves the quality of the learned representations, particularly in overfitting scenarios. Refer to Appendix E.1 for the complete algorithm pseudo code.

### 4.1 STRUCTURE-PRESERVING FRAMEWORK

**Local Structure Context Preservation.** Local structural information characterizes the short-range context dependencies among nodes in the graph. To preserve it, we maintain the ability of mutual inference between a node and its contextual neighbors. The mutual inferability is quantified by the mutual information (MI) $\mathcal{I}_\phi(\tilde{Z}; Z)$, where $\tilde{Z}$ is the random variable of neighborhood embeddings $\tilde{\mathbf{Z}}$, and $Z$ is the random variable of anchor node embeddings $\mathbf{Z}$. Let $P(Z, \tilde{Z})$ be the joint distribution with $P(Z)$ and $P(\tilde{Z})$ the marginal distributions. We apply the Jensen-Shannon MI estimator to maximize $\mathcal{I}_\phi(\tilde{Z}; Z)$ as follow:

$$\mathcal{I}_{\mathcal{D},\phi}^{JSD}(\tilde{Z}; Z) := \mathbb{E}_{(z,\tilde{z}) \sim P(Z,\tilde{Z})} \log \mathcal{D}(z, \tilde{z}) + \mathbb{E}_{z \sim P(Z), \tilde{z} \sim P(\tilde{Z})} \log(1 - \mathcal{D}(z, \tilde{z})). \tag{5}$$

For the optimal discriminator $\mathcal{D}^*$, $\mathcal{I}_{\mathcal{D}^*,\phi}^{JSD}(\tilde{Z}; Z) = 2D_{\text{JS}}(P(Z, \tilde{Z}) \| P(Z)P(\tilde{Z})) - \log 4$ (see Appendix C for a proof). Therefore, any parameters that maximize the above estimator also maximize the JS divergence between the joint and marginal distributions. Moreover, as shown in (Hjelm et al., 2018), $D_{\text{JS}}(P(Z, \tilde{Z}) \| P(Z)P(\tilde{Z}))$ is a monotonic function of the point-wise mutual information, which implies that maximizing it is equivalent to maximize the mutual information $\mathcal{I}_\phi(\tilde{Z}; Z)$.

Therefore, we take maximizing $\mathcal{I}_{\mathcal{D},\phi}^{JSD}(\tilde{Z}; Z)$ as the optimization objective, and rewrite it to obtain the empirical objective as follows:

$$\begin{aligned}
&\arg\max_{\mathcal{D},\phi} \left\{ \mathcal{I}_{\mathcal{D},\phi}^{JSD}(\tilde{Z}; Z) \right\} \\
&= \arg\max_{\mathcal{D},\phi} \left\{ \mathbb{E}_{(z,\tilde{z}) \sim P(Z,\tilde{Z})} \log \mathcal{D}(z, \tilde{z}) + \mathbb{E}_{z \sim P(Z), \tilde{z} \sim P(\tilde{Z})} \log(1 - \mathcal{D}(z, \tilde{z})) \right\} \\
&= \arg\min_{\mathcal{D},\phi} \left\{ -\mathbb{E}_{\tilde{z} \sim P(\tilde{Z})} \left[ \mathbb{E}_{z \sim P(Z|\tilde{Z})} \log \mathcal{D}(z, \tilde{z}) + \mathbb{E}_{z \sim P(Z)} \log(1 - \mathcal{D}(z, \tilde{z})) \right] \right\} \\
&\approx \arg\min_{\mathcal{D},\phi} \left\{ -\frac{1}{|\mathcal{V}|} \sum_{i \in \mathcal{V}} \left[ \log \mathcal{D}(\mathbf{z}_i, \tilde{\mathbf{z}}_i) + \log(1 - \mathcal{D}(\mathbf{z}_j, \tilde{\mathbf{z}}_i)) \right] \right\},
\end{aligned} \tag{6}$$

where we treat observed node-neighbor context pairs $(\mathbf{z}_i, \tilde{\mathbf{z}}_i)$ as positive samples from joint distribution, and randomly pairing $\tilde{\mathbf{z}}_i$ with the embedding $\mathbf{z}_j$ of a different node $j \neq i$, which approximates a sample from the product of marginals. To obtain the neighbor embedding $\tilde{\mathbf{z}}_i$ for each node $i$, we employ a graph aggregation operator to perform context representation aggregation, which is defined as follow:

$$\tilde{\mathbf{z}}_i = \text{Aggregator}\left(\{\mathbf{z}_k | k \in \mathcal{N}_i \cup \{i\}\}\right). \tag{7}$$

**Global Structure Equivalence Preservation.** In addition to local context, we further require the embeddings to preserve the role of nodes within the whole graph structure. To this end, structurally equivalent nodes (e.g., those with the same centrality) should be encouraged to have similar embeddings. Therefore, we introduce a proxy optimization objective of graph centrality reconstruction, which predicts node centralities to preserve the embeddings' awareness of global structural roles.

Let $C \in \mathbb{R}^{N \times B}$ denote the centrality matrix of a graph with $N$ nodes and $B$ different centrality measures. Each entry $c_{i,j}$ represents the value of the $j$-th centrality measure for node $v_i$. Thus, the $i$-th row $C_{i,:}$ corresponds to the centrality profile of node $v_i$ across all $B$ measures, while the $j$-th column $C_{:,j}$ contains the values of the $j$-th centrality measure for all nodes.

We optimize the reconstruction objective as follow:

$$\arg\min_{\xi,\phi} \left\{ \frac{1}{NB} \|h_\xi(f_\phi(\mathbf{X}, \mathbf{A})) - C\|^2 \right\}, \tag{8}$$

where $h_\xi$ is a proxy centrality predicting network.

**Regularized learning objective.** In summary, the total optimization objective of structure decoder is as follow:

$$\mathcal{L}_{\text{reg}} = -\frac{1}{|\mathcal{V}|} \sum_{i \in \mathcal{V}} \left[ \log \mathcal{D}(\mathbf{z}_i, \tilde{\mathbf{z}}_i) + \log(1 - \mathcal{D}(\mathbf{z}_j, \tilde{\mathbf{z}}_i)) \right] + \frac{1}{NB} \|h_\xi(f_\phi(\mathbf{X}, \mathbf{A})) - C\|^2. \tag{9}$$

The final optimization objective of the regularized GCL is:

$$\mathcal{L} = \mathcal{L}_{\text{con}} + \mathcal{L}_{\text{reg}}, \tag{10}$$

where $\mathcal{L}_{\text{con}}$ is a certain contrastive loss, such as Equation (1).

### 4.2 Post-hoc structural augmentation

Beyond regularizing the optimization process, we introduce an explicit structure injection mechanism to directly enhance node embeddings. As analyzed in Section 3.2, the main source of structural information loss in embeddings is the failure of the encoder's message-passing. Motivated by SGC (Wu et al., 2019), we remedy this by applying message-passing directly on the embeddings to explicitly inject structural information. Let $\text{MP}(\mathbf{X}, \mathbf{A}) = \hat{\mathbf{D}}^{-\frac{1}{2}}\hat{\mathbf{A}}\hat{\mathbf{D}}^{-\frac{1}{2}}\mathbf{X}$ denotes a single layer message-passing rule, where $\hat{\mathbf{A}} = \mathbf{A} + \mathbf{I}_N$ is the adjacency matrix with inserted self-loops, and $\hat{\mathbf{D}}$ is its corresponding degree matrix. The structure-augmented embeddings $\mathbf{Z}_{\text{aug}}$ are defined as follow:

$$\mathbf{Z}_{\text{aug}} = \underbrace{\text{MP}_T \circ \cdots \circ (\text{MP}_2\,(\text{MP}_1\,(\mathbf{Z}, \mathbf{A})\,, \mathbf{A})\,, \cdots, \mathbf{A})}_{T \text{ layers of message-passing}} \tag{11}$$

where $\mathbf{Z} = f_\phi(\mathbf{A}, \mathbf{X})$ is the output embeddings of GNN encoder. Since SPR does not make any assumptions about the contrastive loss, it is a framework that compatible with various GCL methods.

## 5 Experimental Study

### 5.1 Experimental Setup

**Datasets and Baselines**  We conduct experiments on six widely used benchmark datasets: Cora, CiteSeer, PubMed (Yang et al., 2016), Am-Photo, Co-Cs (Shchur et al., 2019), and Wiki-CS (Mernyei & Cangea, 2020). We compare the performance of base GCL methods and their variants regularized by SPR under the same hyper-parameters (following their original designs). Specifically, we adopt eight representative GCL models, including GRACE (Zhu et al., 2020), GCA (Zhu et al., 2021b), PiGCL (He et al., 2024), ReGCL (Ji et al., 2024), ProGCL (Xia et al., 2022), GRACE+ (Chi & Ma, 2024), HomoGCL (Li et al., 2023), and GRAPE (Hao et al., 2024). More details about the datasets and baselines are provided in Appendix D.

**Implementation Details**  We use a bi-linear scoring function $\mathcal{D}\,(\mathbf{h}_i, \mathbf{h}_j) = \sigma\,(\mathbf{h}_i\mathbf{W}^\top\mathbf{h}_j)$ as the discriminator network in Equation (6), where $\mathbf{W}^\top$ is the trainable matrix and $\sigma$ is the sigmoid function, and we use GCN convolution operator as the aggregator in Equation (7) for two-hop context aggregation to construct neighbor embeddings. We use degree, betweenness, average neighbor degree, and PageRank as node centrality measures. For the post-hoc structural augmentation in Equation (11), we set the number of message-passing layers $T = 2$. More implementation details and hyper-parameter settings are provided in Appendix E.2.

**Evaluation Protocol**  To ensure an unbiased evaluation of GCL models, we assess the embeddings extracted from the encoder at the epoch where the contrastive loss has converged, and use them to train a downstream node classifier. All baseline methods are trained for 10,000 iterations with a cosine annealing learning rate scheduler (Loshchilov & Hutter, 2016) to guarantee convergence. For dataset splits, we follow the standard public settings for Cora, CiteSeer, and PubMed (20/50/1000 for train/val/test), and adopt random 10%/10%/80% splits for Co-Cs and Am-Photo, and Wiki-CS follow (Zhu et al., 2021b). The downstream node classifier is implemented as logistic regression (Kleinbaum et al., 2002). We tune it on the validation set to select the best classifier and then evaluate on the test set. For each experiment, we repeat it 10 times using different random seeds, and present the mean and standard deviation of the result accuracy.

### 5.2 Main Results and Ablation

To demonstrate the effect of SPR, as well as the individual contributions of the structure decoder and post-hoc structural augmentation during GCL training, we compare the node classification accuracy of base GCL methods and their regularized variants.

Table 3: Node classification accuracy (%) with converged encoders. Results are reported for variants with post-hoc augmentation (+PA), structure decoder (+SD), and their combination (+SPR). The top 1st, 2nd and 3rd results are highlighted with accuracy improvements. OOM denotes out of memory.

| Method | Cora | CiteSeer | PubMed | Am-Photo | Co-CS | Wiki-CS |
|---|---|---|---|---|---|---|
| GRACE | $51.00_{\pm0.00}$ | $51.68_{\pm0.41}$ | $61.88_{\pm0.86}$ | $90.28_{\pm0.15}$ | $84.90_{\pm0.08}$ | $79.03_{\pm0.05}$ |
| +PA | $66.08_{\pm0.30}15.08\uparrow$ | $55.76_{\pm0.83}4.08\uparrow$ | $68.72_{\pm0.43}6.84\uparrow$ | $92.39_{\pm0.29}2.11\uparrow$ | $86.96_{\pm0.17}2.06\uparrow$ | $79.04_{\pm0.29}0.01\uparrow$ |
| +SD | $74.56_{\pm0.71}23.56\uparrow$ | $66.84_{\pm0.94}15.16\uparrow$ | $79.36_{\pm0.39}17.48\uparrow$ | $90.40_{\pm0.17}0.12\uparrow$ | $88.69_{\pm0.17}4.69\uparrow$ | $79.16_{\pm0.09}0.13\uparrow$ |
| +SPR | $75.12_{\pm0.52}24.12\uparrow$ | $64.80_{\pm0.52}13.12\uparrow$ | $80.20_{\pm0.28}18.32\uparrow$ | $92.67_{\pm0.35}2.39\uparrow$ | $87.82_{\pm0.27}2.92\uparrow$ | $79.40_{\pm0.53}0.37\uparrow$ |
| GCA | $64.12_{\pm0.94}$ | $42.26_{\pm0.73}$ | $54.42_{\pm1.54}$ | $90.39_{\pm0.02}$ | $79.45_{\pm0.03}$ | $71.91_{\pm0.06}$ |
| +PA | $70.46_{\pm1.27}6.34\uparrow$ | $46.14_{\pm0.61}3.88\uparrow$ | $62.38_{\pm0.99}7.96\uparrow$ | $90.75_{\pm0.02}0.36\uparrow$ | $85.62_{\pm0.03}6.17\uparrow$ | $72.73_{\pm0.08}0.82\uparrow$ |
| +SD | $72.50_{\pm1.11}8.38\uparrow$ | $55.10_{\pm0.55}12.84\uparrow$ | $65.50_{\pm2.26}11.08\uparrow$ | $92.52_{\pm0.26}2.13\uparrow$ | $89.13_{\pm0.07}9.68\uparrow$ | $79.82_{\pm0.07}7.91\uparrow$ |
| +SPR | $75.38_{\pm0.87}11.26\uparrow$ | $57.40_{\pm1.03}15.14\uparrow$ | $65.82_{\pm0.93}11.40\uparrow$ | $91.82_{\pm0.08}1.43\uparrow$ | $89.50_{\pm0.04}10.05\uparrow$ | $79.60_{\pm0.09}7.69\uparrow$ |
| PiGCL | $60.22_{\pm0.41}$ | $44.18_{\pm0.68}$ | $55.36_{\pm1.87}$ | $90.54_{\pm0.03}$ | $80.39_{\pm0.04}$ | $77.32_{\pm0.03}$ |
| +PA | $62.10_{\pm1.22}1.88\uparrow$ | $46.92_{\pm0.48}2.74\uparrow$ | $68.06_{\pm0.40}12.70\uparrow$ | $91.72_{\pm0.07}1.18\uparrow$ | $84.68_{\pm0.01}4.29\uparrow$ | $77.55_{\pm0.06}0.22\uparrow$ |
| +SD | $62.14_{\pm1.31}1.92\uparrow$ | $47.92_{\pm0.98}3.74\uparrow$ | $79.02_{\pm0.22}23.66\uparrow$ | $93.29_{\pm0.06}2.75\uparrow$ | $88.87_{\pm0.11}8.48\uparrow$ | $80.33_{\pm0.28}3.01\uparrow$ |
| +SPR | $68.42_{\pm0.20}8.20\uparrow$ | $51.04_{\pm0.55}6.86\uparrow$ | $79.26_{\pm0.88}23.90\uparrow$ | $92.59_{\pm0.26}2.05\uparrow$ | $89.31_{\pm0.07}8.92\uparrow$ | $80.00_{\pm0.14}2.68\uparrow$ |
| ReGCL | $52.02_{\pm0.64}$ | $43.26_{\pm0.72}$ | OOM | OOM | OOM | OOM |
| +PA | $64.14_{\pm0.79}12.12\uparrow$ | $52.02_{\pm1.33}8.76\uparrow$ | N/A | N/A | N/A | N/A |
| +SD | $68.68_{\pm0.53}16.66\uparrow$ | $54.36_{\pm0.94}11.10\uparrow$ | N/A | N/A | N/A | N/A |
| +SPR | $71.62_{\pm0.50}19.60\uparrow$ | $57.54_{\pm1.40}14.28\uparrow$ | N/A | N/A | N/A | N/A |
| ProGCL | $65.40_{\pm1.14}$ | $47.00_{\pm0.67}$ | $63.26_{\pm1.26}$ | $91.70_{\pm0.03}$ | $80.20_{\pm0.01}$ | $76.94_{\pm0.01}$ |
| +PA | $72.32_{\pm0.87}6.92\uparrow$ | $53.10_{\pm1.07}6.10\uparrow$ | $68.66_{\pm0.74}5.40\uparrow$ | $91.73_{\pm0.10}0.03\uparrow$ | $84.85_{\pm0.04}4.65\uparrow$ | $77.26_{\pm0.06}0.32\uparrow$ |
| +SD | $72.24_{\pm0.53}6.84\uparrow$ | $59.06_{\pm0.56}12.06\uparrow$ | $77.82_{\pm0.45}14.56\uparrow$ | $92.18_{\pm0.21}0.48\uparrow$ | $91.25_{\pm0.10}11.05\uparrow$ | $79.25_{\pm0.36}2.31\uparrow$ |
| +SPR | $77.52_{\pm0.57}12.12\uparrow$ | $60.22_{\pm0.98}13.22\uparrow$ | $79.20_{\pm0.53}15.94\uparrow$ | $92.91_{\pm0.12}1.21\uparrow$ | $90.69_{\pm0.04}10.49\uparrow$ | $78.98_{\pm0.29}2.04\uparrow$ |
| GRACE+ | $71.64_{\pm2.48}$ | $62.48_{\pm0.89}$ | $75.10_{\pm0.94}$ | OOM | OOM | OOM |
| +PA | $72.98_{\pm2.64}1.34\uparrow$ | $62.72_{\pm0.49}0.24\uparrow$ | $76.23_{\pm0.04}1.13\uparrow$ | N/A | N/A | N/A |
| +SD | $71.68_{\pm0.65}0.04\uparrow$ | $62.96_{\pm0.67}0.48\uparrow$ | $78.43_{\pm0.92}3.33\uparrow$ | N/A | N/A | N/A |
| +SPR | $73.44_{\pm0.65}1.80\uparrow$ | $63.02_{\pm1.12}0.54\uparrow$ | $78.83_{\pm0.21}3.73\uparrow$ | N/A | N/A | N/A |
| HomoGCL | $69.64_{\pm0.08}$ | $46.52_{\pm0.92}$ | $70.24_{\pm0.62}$ | $92.89_{\pm0.29}$ | $89.27_{\pm0.40}$ | $79.03_{\pm0.07}$ |
| +PA | $72.84_{\pm1.22}3.20\uparrow$ | $49.64_{\pm0.95}3.12\uparrow$ | $76.12_{\pm0.52}5.88\uparrow$ | $88.81_{\pm0.27}$ | $85.26_{\pm0.53}$ | $76.45_{\pm0.81}$ |
| +SD | $75.00_{\pm0.18}5.36\uparrow$ | $58.44_{\pm0.15}11.92\uparrow$ | $78.12_{\pm0.52}7.88\uparrow$ | $91.06_{\pm0.18}$ | $90.04_{\pm0.10}0.77\uparrow$ | $81.26_{\pm0.34}2.23\uparrow$ |
| +SPR | $75.64_{\pm0.34}6.00\uparrow$ | $59.68_{\pm1.03}13.16\uparrow$ | $76.68_{\pm0.24}6.44\uparrow$ | $89.70_{\pm0.27}$ | $87.94_{\pm0.22}$ | $78.38_{\pm0.41}$ |
| GRAPE | $57.92_{\pm0.10}$ | $48.56_{\pm1.35}$ | $68.76_{\pm0.20}$ | $92.71_{\pm0.17}$ | $84.9_{\pm0.03}$ | $81.86_{\pm0.14}$ |
| +PA | $66.88_{\pm0.45}8.96\uparrow$ | $50.64_{\pm0.79}2.08\uparrow$ | $73.76_{\pm0.71}5.00\uparrow$ | $91.84_{\pm0.49}$ | $86.89_{\pm0.04}1.99\uparrow$ | $78.70_{\pm0.42}$ |
| +SD | $78.96_{\pm0.73}21.04\uparrow$ | $66.68_{\pm0.48}18.12\uparrow$ | $76.04_{\pm0.32}7.28\uparrow$ | $91.97_{\pm0.06}$ | $89.48_{\pm0.12}4.49\uparrow$ | $79.91_{\pm0.17}$ |
| +SPR | $77.60_{\pm0.59}19.68\uparrow$ | $67.44_{\pm1.18}18.88\uparrow$ | $75.36_{\pm1.08}6.60\uparrow$ | $90.07_{\pm0.47}$ | $88.85_{\pm0.24}3.95\uparrow$ | $78.80_{\pm0.24}$ |

As shown in Table 3, **firstly**, we observe that SPR can improve the accuracy gained by baseline methods in most conditions after seamlessly integrated into them in a plug-and-play manner. This suggests that SPR effectively mitigates the overfitting issue present in existing GCL approaches. **Secondly**, the accuracy gains are more pronounced for structure-sensitive datasets (Cora, CiteSeer, and PubMed) than for structure-insensitive datasets (Am-Photo, Co-CS, and Wiki-CS). This results further supports our analysis in Section 3 for datasets that are inherently less dependent on structural information, the loss of such information has a limited impact on the encoder's performance in downstream classification. **Thirdly**, among the baseline methods, GRACE+ and HomoGCL achieves relative better converged accuracy, this can be attributed to their pre-designed structure-aware contrastive loss. GRACE+ estimates node similarity and samples negatives from a small set of high-confidence nodes based on prior graph structure, thereby incorporating structural information. HomoGCL similarly augments the positive set using homophily of graph. These approaches align with our proposed idea that GCL should preserve structural information in encoders.

We also conduct ablation experiment by applying structure decoder (SD) and post-hoc structural augmentation (PA) individually to the baseline methods, the results of which are shown in Table 3. For structure-sensitive datasets, embeddings enhanced with PA almost always improve accuracy compared to the original learned embeddings (e.g., GRACE vs. GRACE+PA, GRACE+SD vs. GRACE+SPR), indicating that PA is an effective embedding augmentation strategy. In contrast, for structure-insensitive datasets, introducing PA may lead to over-smoothing of node embeddings, slightly reducing accuracy (e.g., HomoGCL on Am-Photo, Co-CS, and Wiki-CS), this phenomenon is typically observed when the original embeddings already achieve high accuracy. Meanwhile, SD yields consistent accuracy improvements across various datasets and baseline methods, which show it effectiveness. However, SPR produces a better overall performance than that of AP or SD alone, benefiting from the combination of them.

## 5.3 EMBEDDING VISUALIZATION

In this section, we visualize node embeddings during training. Firstly, based on label information, we divide the negative set of anchor nodes during training into two subsets: true negatives (with labels different from the anchor node) and false negatives (with the same label as the anchor node). We then

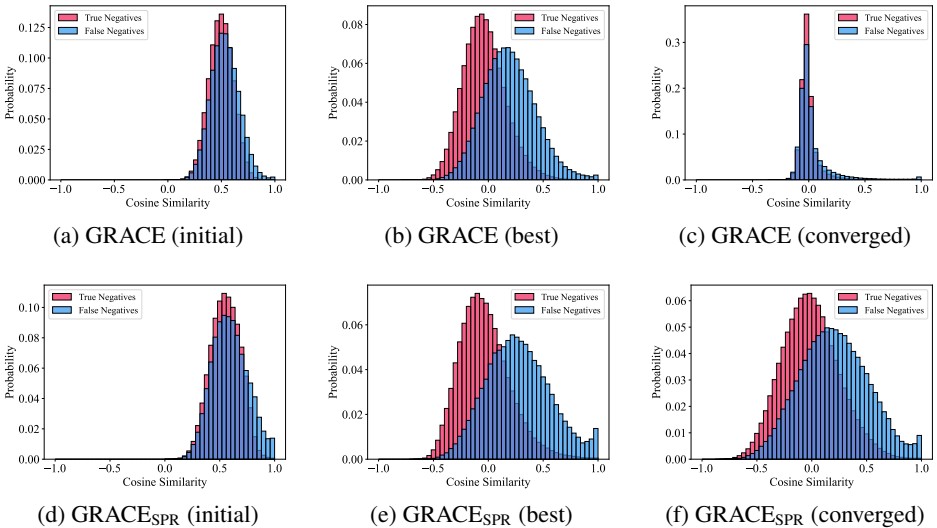

(a) GRACE (initial)  (b) GRACE (best)  (c) GRACE (converged)

(d) GRACE$_{SPR}$ (initial)  (e) GRACE$_{SPR}$ (best)  (f) GRACE$_{SPR}$ (converged)

Figure 4: The similarity distributions between anchors and true negatives, as well as between anchors and false negatives. The results are obtained on CiteSeer dataset.

visualize the similarity distributions between anchor nodes and true negatives, as well as that between anchor nodes and false negatives. Ideally, these two distributions should form a non-overlapping bimodal pattern, with the expected similarity of true negatives being lower than that of false negatives. Figure 4 illustrates the similarity distributions of GRACE and GRACE$_{+SPR}$ at the early, best, and final stages of training. We observe that, without regularization, GRACE will push both true and false negatives away during training, while GRACE$_{+SPR}$ consistently preserves the bimodal distribution.

We further visualize the impact of PA on node embeddings through t-SNE dimensionality reduction. As shown in Figure 5, after a simple parameter-free message-passing, the quality of node embeddings can be clearly improved. This demonstrates the effectiveness of our post-hoc augmentation, especially under overfitting scenarios. More visualization results, such as intra-class similarity and comparisons of classification accuracy curves during training, are presented in Appendix B.



Figure 5: node embedding t-SNE visualization.

## 6 CONCLUSIONS

This paper revisits Graph Contrastive Learning (GCL) through the lens of contrastive overfitting. We highlight a critical yet previously overlooked issue: empirically optimal GCL encoders often lead to poor downstream performance. Our analysis reveals that this overfitting arises from the structure-agnostic nature of the contrastive loss, which results in the loss of essential graph structural information. To mitigate this problem, we propose a simple yet effective Structure-Preserving Regularization (SPR) approach that introduces structural priors by preserving both the mutual inferability between a node and its neighborhood as well as its centrality reconstruction ability. This work sheds new light on the generalization behavior of GCL and provides a practical path toward building more reliable unsupervised graph learning frameworks.

## 7 LLM USAGE AND REPRODUCIBILITY

This manuscript has been polished with the assistance of a ChatGPT. The authors take full responsibility for the content. This work adheres to the ICLR Code of Ethics. To facilitate reproducibility, our anonymous code implementation is available at: https://anonymous.4open.science/r/SPR-GCL-DDB3/

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

# A  RELATED WORKS

**Graph Neural Network**  Graph Neural Networks (GNNs) (Kipf & Welling, 2017; Veličković et al., 2018; Hamilton et al., 2017; Wu et al., 2020) have become a fundamental architecture for learning representations from graph-structured data. Most widely used GNN layers are built upon the message-passing mechanism (Gilmer et al., 2017), which iteratively aggregates information from a node's neighbors to encode the structural properties of the graph.

**Graph Contrastive Learning**  Contrastive learning (CL) (Oord et al., 2018; Chen et al., 2020; He et al., 2020; Zbontar et al., 2021) has emerged as a prominent self-supervised learning paradigm that captures the inherent similarities and differences among data instances, thereby reducing the dependence on labeled data. Its core principle is to pull together representations of similar instances (positive pairs) while pushing apart those of dissimilar instances (negative pairs) in the embedding space. There are many work adapting CL to graph representation learning, known as graph contrastive learning (GCL) (Veličković et al., 2018; Zhang et al., 2021; Zhu et al., 2020; Hassani & Khasahmadi, 2020; Zhu et al., 2021a), which brings a new paradigm in self-supervised graph representation learning.

**False Negatives in GCL**  In GCL methods based on the InfoNCE loss, one prominent issue is that all other nodes, apart from the anchor itself, are treated as negative samples and are pushed away in the embedding space. Several studies have pointed out that many of these negatives are in fact false negatives-nodes sharing the same class label as the anchor-which ideally should not be repelled. To address the issue of imprecise positive and negative sample sets, various approaches have been proposed Xia et al. (2022); Li et al. (2023); Chi & Ma (2024); Hao et al. (2024). For instance, Xia et al. (2022) employs a mixture of Beta distributions to estimate the likelihood of a node being a false negative. However, these methods generally assume access to implicit label-related signals and focus on improving the theoretical upper bound of GCL performance, often neglecting the behavior of encoders at convergence. Existing works also tend to treat label information as a monolithic entity, without distinguishing between structural and attribute-based components. Furthermore, most studies evaluate their methods on datasets where label dependence on structure varies, but the implications of this factor are seldom explored.

# B  ADDITIONAL EXPERIMENTAL RESULTS

## B.1  CONTRASTIVE OVERFITTING

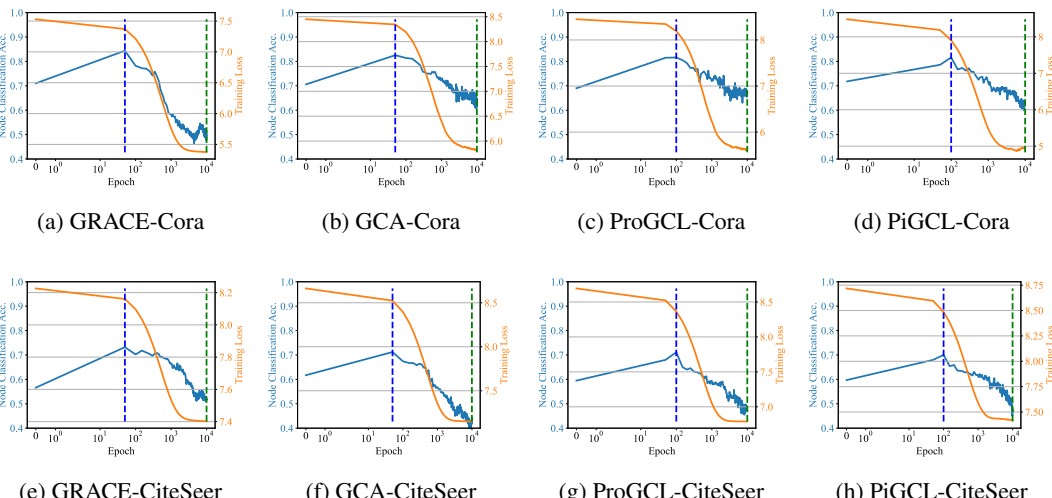

(a) GRACE-Cora  (b) GCA-Cora  (c) ProGCL-Cora  (d) PiGCL-Cora

(e) GRACE-CiteSeer  (f) GCA-CiteSeer  (g) ProGCL-CiteSeer  (h) PiGCL-CiteSeer

Figure 6: GCL loss and node classification accuracy over training epochs.

Figure 6 illustrates the widespread presence of contrastive overfitting across different GCL methods and datasets.

## B.2    PERFORMANCE DEGRADATION

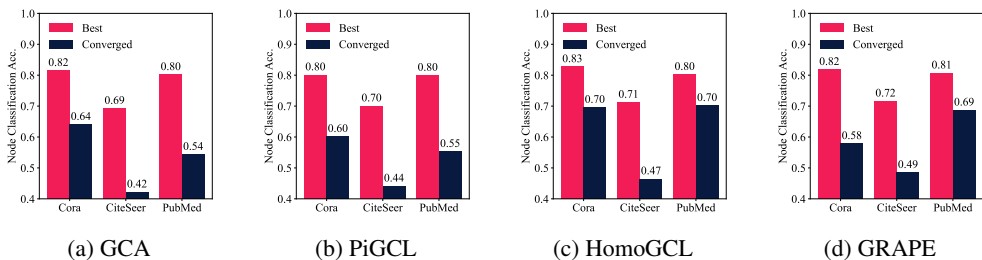

Figure 7: Performance degradation of GCA, PiGCL, HomoGCL, and GRAPE on different datasets.

Figure 7 shows the performance degradation of four InfoNCE-based GCL methods.

## B.3    EXTRA VISUALIZATION

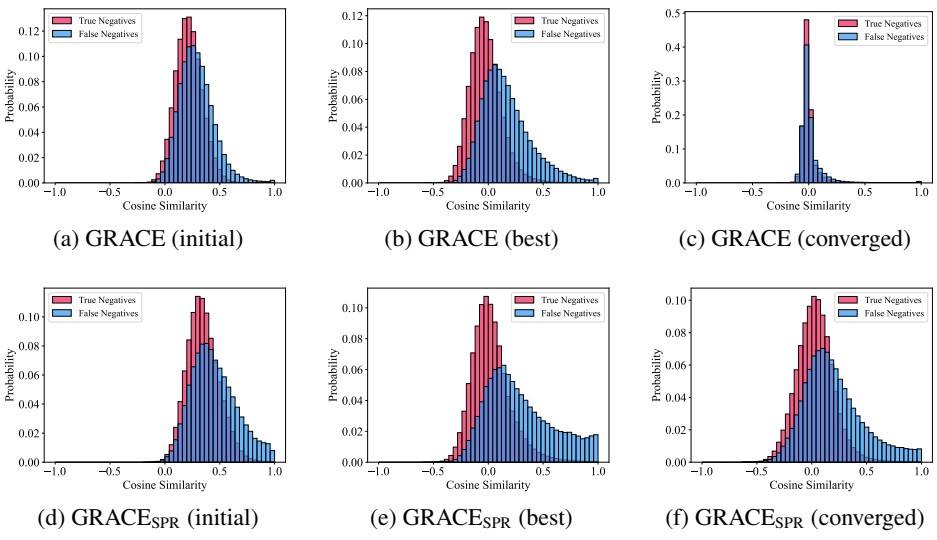

Figure 8: The similarity distributions between anchors and true negatives, as well as between anchors and false negatives. The results are obtained on Cora dataset.

Figure 8 shows the similarity distributions of GRACE and GRACE$_{+SPR}$ at the early, best, and final stages of training.

Figure 9 shows that after applying the SPR regularization strategy, the downward trend in downstream task accuracy during training is significantly alleviated.

Figures 10 to 12 illustrate the intra-class node similarity. By comparison, we can observe that SPR effectively preserves the similarity among nodes of the same class at convergence.

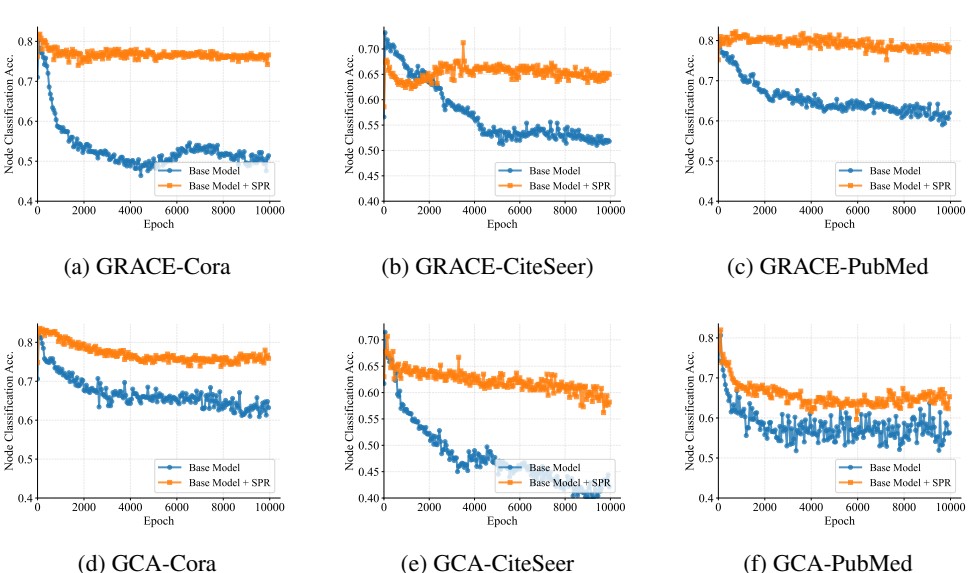

(a) GRACE-Cora      (b) GRACE-CiteSeer)      (c) GRACE-PubMed

(d) GCA-Cora      (e) GCA-CiteSeer      (f) GCA-PubMed

Figure 9: Comparison of downstream node classification accuracy during training.

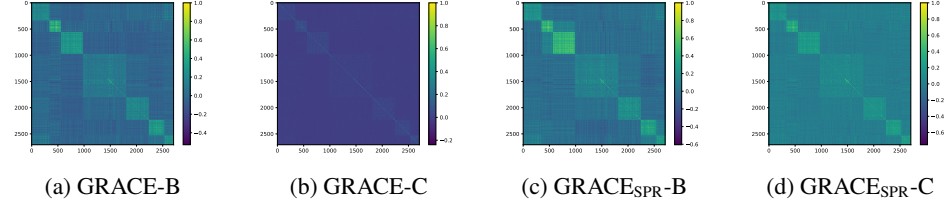

(a) GRACE-B      (b) GRACE-C      (c) GRACE$_{SPR}$-B      (d) GRACE$_{SPR}$-C

Figure 10: Intra-class node similarity matrix on the Cora dataset, with nodes (rows and columns) reordered by class. "B" represent Best, "C" represent Converged.



(a) GRACE-B      (b) GRACE-C      (c) GRACE$_{SPR}$-B      (d) GRACE$_{SPR}$-C

Figure 11: Intra-class node similarity matrix on the CiteSeer dataset, with nodes (rows and columns) reordered by class. "B" represent Best, "C" represent Converged.

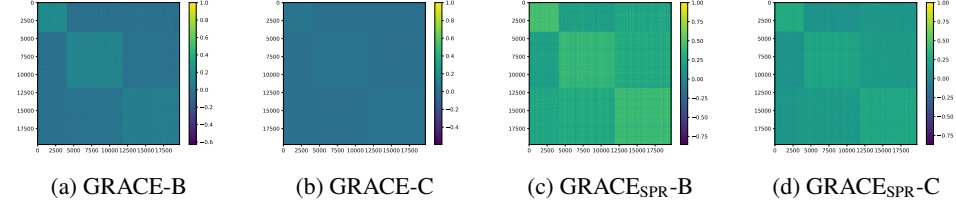

(a) GRACE-B      (b) GRACE-C      (c) GRACE$_{SPR}$-B      (d) GRACE$_{SPR}$-C

Figure 12: Intra-class node similarity matrix on the PubMed dataset, with nodes (rows and columns) reordered by class. "B" represent Best, "C" represent Converged.

## C   PROOFS

**Theorem 1.** *Let $\mathcal{D}$ be a discriminator trained to distinguish between joint samples $(z, \tilde{z}) \sim P(Z, \tilde{Z})$ and marginal samples $(z, \tilde{z}) \sim P(Z)P(\tilde{Z})$. Consider the following objective:*

$$\mathcal{I}_{\mathcal{D}}^{\mathrm{JSD}}(\tilde{Z}; Z) := \mathbb{E}_{(z,\tilde{z}) \sim P(Z,\tilde{Z})} \left[ \log \mathcal{D}(z, \tilde{z}) \right] + \mathbb{E}_{z \sim P(Z), \tilde{z} \sim P(\tilde{Z})} \left[ \log(1 - \mathcal{D}(z, \tilde{z})) \right].$$

*At the optimal discriminator $\mathcal{D}^*$, the objective evaluates to:*

$$\mathcal{I}_{\mathcal{D}^*}^{\mathrm{JSD}}(\tilde{Z}; Z) = 2 D_{\mathrm{JS}} \left( P(Z, \tilde{Z}) \, \| \, P(Z)P(\tilde{Z}) \right) - \log 4,$$

*where $D_{\mathrm{JS}}(\cdot \| \cdot)$ denotes the Jensen-Shannon divergence.*

*Proof.* We rewrite the term $\mathcal{I}_{\mathcal{D}}^{\mathrm{JSD}}(\tilde{Z}; Z)$ as:

$$\begin{aligned}
\mathcal{I}_{\mathcal{D}}^{\mathrm{JSD}}(\tilde{Z}; Z) &:= \mathbb{E}_{(z,\tilde{z}) \sim P(Z,\tilde{Z})} \left[ \log \mathcal{D}(z, \tilde{z}) \right] + \mathbb{E}_{z \sim P(Z), \tilde{z} \sim P(\tilde{Z})} \left[ \log(1 - \mathcal{D}(z, \tilde{z})) \right] \\
&= \int p(z, \tilde{z}) \log(\mathcal{D}(z, \tilde{z})) + p(z)p(\tilde{z}) \log(1 - \mathcal{D}(z, \tilde{z})) dz d\tilde{z},
\end{aligned} \quad (12)$$

since $\mathcal{I}_{\mathcal{D}}^{\mathrm{JSD}}(\tilde{Z}; Z)$ is concave in $\mathcal{D}$, we calculate the first-order derivative:

$$\frac{\partial \mathcal{I}_{\mathcal{D}}^{\mathrm{JSD}}(\tilde{Z}; Z)}{\partial \mathcal{D}} = \int \frac{p(z, \tilde{z})}{\mathcal{D}} - \frac{p(z)p(\tilde{z})}{1 - \mathcal{D}} dz d\tilde{z},$$

let $\frac{\partial \mathcal{I}_{\mathcal{D}}^{\mathrm{JSD}}(\tilde{Z}; Z)}{\partial \mathcal{D}} = 0$, we have $\mathcal{D}^* = \frac{p(z, \tilde{z})}{p(z, \tilde{z}) + p(z)p(\tilde{z})}$.

We plugging $\mathcal{D}^*$ back to Equation (12):

$$\mathcal{I}_{\mathcal{D}^*}^{\mathrm{JSD}}(\tilde{Z}; Z) = \int p(z, \tilde{z}) \log[\frac{p(z, \tilde{z})}{p(z, \tilde{z}) + p(z)p(\tilde{z})}] + p(z)p(\tilde{z}) \log[\frac{p(z)p(\tilde{z})}{p(z, \tilde{z}) + p(z)p(\tilde{z})}] dz d\tilde{z} \quad (13)$$

Let $p$ denotes $p(z, \tilde{z})$, $q$ denotes $p(z)p(\tilde{z})$,

$$\begin{aligned}
\mathcal{I}_{\mathcal{D}^*}^{\mathrm{JSD}}(\tilde{Z}; Z) &= \int p \log[\frac{p}{p + q}] + q \log[\frac{q}{p + q}] dz d\tilde{z} \\
&= \int p \log[\frac{p}{p + q}] + p \log(2) + q \log[\frac{q}{p + q}] + q \log(2) - p \log(2) - q \log(2) dz d\tilde{z} \\
&= \int p \log[\frac{2p}{p + q}] + q \log[\frac{2q}{p + q}] dz d\tilde{z} - (\int p dz d\tilde{z} + \int q dz d\tilde{z}) \cdot \log(2) \\
&= 2 D_{\mathrm{JS}}(p \| q) - \log(4) \\
&= 2 D_{\mathrm{JS}} \left( P(Z, \tilde{Z}) \, \| \, P(Z)P(\tilde{Z}) \right) - \log 4
\end{aligned} \quad (14)$$

$\square$

## D   BASELINES AND DATASETS

### D.1   BASELINES

In this section, we give brief introductions of the baselines used in the paper which are not described in the main paper due to the space constraint.

- **GRACE** learns node representations by generating two graph views (edge dropping + feature masking) and maximizing their agreement based on InfoNCE loss Chen et al. (2020). Code link: https://github.com/CRIPAC-DIG/GRACE
- **GCA** performs adaptive augmentation that drops unimportant edges and perturbs unimportant features based on centrality. Code link: https://github.com/CRIPAC-DIG/GCA

- **PiGCL** detects embedding-and-ignoring conflicts via gradient cues and dynamically ignores those negatives during training so the encoder can learn from them adaptively. Code link: `https://github.com/hedongxiao-tju/PiGCL`

- **ReGCL** addresses GNN-GCL conflicts through gradient-guided structure learning and gradient-weighted InfoNCE. Code link: `https://github.com/RingBDStack/ReGCL`

- **ProGCL** models the distribution of negative pairs using a Beta Mixture Model (BMM), enabling it to estimate the probability of a negative sample being a false negative based on embedding similarity. Code link: `https://github.com/junxia97/ProGCL`

- **GRACE Plus**: exploits node similarity to construct anchor-aware sampling distributions which estimates node similarity and samples negatives from a small set of high-confidence nodes. Code link: `https://github.com/frankhlchi/SimEnhancedGCL`

- **HomoGCL**: adopts the homophily assumption by treating all neighbors of an anchor node as positive samples and assigning weights using clustering techniques. Code link: `https://github.com/wenzhilics/HomoGCL`

- **GRAPE**: leverages a subspace-preserving technique to learn the weights of negative samples. Code link: `https://github.com/zz-haooo/WWW24-GRAPE`

## D.2  DATASETS

In this section, we give brief introductions of the datasets used in the paper, Table 4 shows detailed information of each dataset.

Table 4: Dataset information statistics.

| Dataset | #Nodes | #Edges | #Attributes | #Classes |
|---------|--------|--------|-------------|----------|
| Cora | 2,708 | 10,556 | 1,433 | 7 |
| CiteSeer | 3,327 | 9,228 | 3,703 | 6 |
| PubMed | 19,717 | 88,651 | 500 | 3 |
| Co-Cs | 18,333 | 163,788 | 6,805 | 15 |
| Am-Photo | 7,650 | 238,163 | 745 | 8 |
| Wiki-CS | 11,701 | 431,726 | 300 | 10 |

- **Cora** Yang et al. (2016): A citation network where each node represents a scientific publication in the field of machine learning, and edges denote citation relationships between papers. Each publication is described by a sparse bag-of-words feature vector derived from its abstract, and is categorized into one of seven predefined research topics.

- **CiteSeer** Yang et al. (2016): A citation network composed of scientific publications in the field of computer science. Similar to Cora, nodes represent documents and edges represent citation links. Each document is represented by a sparse bag-of-words vector of its content.

- **PubMed** Yang et al. (2016): A citation network of biomedical research papers from the PubMed database. Each node corresponds to a paper, and edges indicate citation links. Node features are TF-IDF weighted word vectors based on the paper abstracts.

- **Co-CS** Shchur et al. (2018): An academic network constructed from the Microsoft Academic Graph, where nodes represent authors and edges denote co-authorship relationships-i.e., two authors are connected if they have collaborated on at least one paper. Each node is associated with a sparse bag-of-words feature vector derived from the keywords of the papers authored by that individual. The label assigned to each author corresponds to their most active research area.

- **Am-Photo** Shchur et al. (2018): A network of co-purchase relationships constructed from Amazon, where nodes represent products and edges indicate that two products are frequently bought together. Each node is associated with a sparse bag-of-words feature vector derived from product reviews and is labeled according to its category.

- **Wiki-CS** Mernyei & Cangea (2020): A reference network derived from Wikipedia, where nodes represent computer science-related articles and edges denote hyperlinks between them. Each node is assigned one of ten class labels, corresponding to distinct subfields within computer science. Node features are computed by averaging the pre-trained GloVe word embeddings of the words appearing in the respective article.

# E IMPLEMENTATION DETAILS

## E.1 ALGORITHM PSEUDO CODE

We provide the algorithm pseudo code as follow:

---

**Algorithm 1** Regularized GCL

---

**Input:** original graph $\mathcal{G} = (\mathbf{A}, \mathbf{X})$, encoder $f_\phi$;
**Output:** the converged encoder $f_{\phi^*}$, node embeddings $\mathbf{Z}$;
1: Initialize encoder $f_\phi$
2: **while** not converge **do**
3:     generate two augmented graph views $\mathcal{G}_U$ and $\mathcal{G}_V$
4:     obtain node embeddings $\mathbf{Z}, \mathbf{U}, \mathbf{V}$ of $\mathcal{G}, \mathcal{G}_U, \mathcal{G}_V$ using encoder $\phi$
5:     compute the contrastive loss $\mathcal{L}_{\text{con}}$ of a base GCL method (e.g., Equation (1))
6:     compute neighbor context embedding $\tilde{\mathbf{Z}}$ by Equation (7)
7:     compute local context mutual inference loss $\mathcal{L}_{\text{MI}}$ by Equation (6)
8:     compute global equivalence loss $\mathcal{L}_{\text{ge}}$ by Equation (8)
9:     the final loss $\mathcal{L} = \mathcal{L}_{\text{con}} + \mathcal{L}_{\text{MI}} + \mathcal{L}_{\text{ge}}$
10:     update the parameters of $f_\phi$ via minimizing $\mathcal{L}$
11: **end while**
12: $\widetilde{\mathbf{A}} \leftarrow \hat{\mathbf{D}}^{-\frac{1}{2}} \hat{\mathbf{A}} \hat{\mathbf{D}}^{-\frac{1}{2}}$
13: $\mathbf{Z} \leftarrow f_{\phi^*}(\mathbf{A}, \mathbf{X})$
14: $\mathbf{Z} \leftarrow \widetilde{\mathbf{A}}^2 \mathbf{Z}$

---

## E.2 IMPLEMENTATION DETAILS

To identify node structural equivalence, we select four important centrality-related node properties to represent a node's role in the graph topology and reconstruct them through decoding. These attributes include node degree, betweenness centrality, average neighbor degree, and PageRank. Degree measures the number of a node's direct connections. Nodes with higher degree centrality are regarded as more locally important, as they engage in more direct interactions within the network. Betweenness measures how often a node appears on the shortest paths between other pairs of nodes. Nodes with high betweenness centrality serve as critical bridges for information flow across the network. PageRank assesses a node's importance based on both the quantity and quality of its incoming connections, assigning higher weight to links from more influential nodes.

For the hyper-parameters of baseline methods, we follow the default settings provided in the official implementations (refer to Appendix D.1). For datasets or methods where hyper-parameters are not specified, we perform a small-scale grid search to approximate the performance reported in the original papers as closely as possible. The searched ranges include: hidden dimension $N_{\text{hid}} \in \{256, 512\}$, edge/feature masking probability $p \in \{0.1, 0.2, 0.4\}$, learning rate $lr \in \{0.001, 0.0001\}$ with a cosine annealing scheduler, temperature coefficient $\tau \in \{0.2, 0.3, 0.5\}$, and projection head dimension $N_{\text{proj}} \in \{256, 512\}$, weight decay rate $\lambda = 1e - 5$, for downstream classifier, $lr = 0.01$, epochs $= 2000$, weight_decay $= 5e - 4$.

All experiments are conducted on an NVIDIA RTX 3090Ti GPU.

