# OpenReview forum: "Revisiting Graph Contrastive Learning through the Lens of Contrastive Overfitting"
_ICLR.cc/2026/Conference — ICLR 2026 Conference Withdrawn Submission_

### Official Review · Reviewer_gwWU · 2025-10-25

**Soundness:** 2
**Presentation:** 3
**Contribution:** 2
**Rating:** 4
**Confidence:** 4

**Summary:**

The paper focuses on the overfitting issue in InfoNCE-based graph contrastive learning caused by the structure-agnostic of the contrastive objective. It proposes a structure-preserving regularization method to mitigate contrastive overfitting by establishing preservation objectives for both global and local structures, along with an explicit structure injection mechanism. Experimental evaluation is conducted on six representative datasets, with comprehensive comparisons against eight baseline methods.

**Strengths:**

1. The paper is well-motivated, with a clear focus on performance degradation in GCL attributed to overfitting induced by the contrastive objective.
2. Empirical observations and experiments are effectively illustrated with compelling visual evidence.
3. The work offers insightful analysis linking overfitting to the structure-agnostic nature of CL, advancing understanding of GCL generalization.

**Weaknesses:**

1. The method is applicable only in cases where InfoNCE-based GCL suffers from severe overfitting on graphs with strong structural dependency, which limits its practical utility and generalizability.
2. Observations 2 and 3 do not generalize to all GCL scenarios, weakening the universality of the analysis.
3. Equations (5–7) are directly derived from DGI (GCL method in 2019) [1], and similar loss-based improvements already exist (e.g., GGD[2]), making the integration of DGI into InfoNCE-based GCL insufficiently novel.
4. The design in Section 4.2 appears more like a heuristic trick, potentially masking issues that should be fundamentally addressed by the core method.
5. The overfitting mitigation is marginal, as the model fails to outperform non-overfitting baselines (e.g., DGI), indicating limited regularization capacity and insufficient analytical depth.
6. The training procedures and implementation details for critical components such as the MI estimator, centrality predictor, and neighborhood aggregator are inadequately specified.

[1]. Veličković, Petar, et al. "Deep Graph Infomax." ICLR 2019.
[2]. Zheng, Yizhen, et al. "Rethinking and scaling up graph contrastive learning: An extremely efficient approach with group discrimination." NeurIPS 2022.

**Questions:**

1. Does overfitting only not occur in DGI-like methods? What about other approaches such as BGRL[3] or MVGRL[4]?
2. In Table 3, accuracy on some datasets is significantly lower than that of DGI. Moreover, under standard settings (e.g., epochs < 1000), GCL typically achieves much higher performance even with suboptimal hyperparameters. How is the validity of Table 3 justified?
3. Equation (8) appears highly intuitive. How is its effectiveness theoretically or empirically supported?
4. The purpose of Figure 4 is unclear. What is the intended meaning behind illustrating true/false negatives, and how does this relate to overfitting?
5. GGD further optimizes the DGI loss (e.g., by improving the summary vector). Why does this work still rely on DGI for design? Does GGD also suffer from overfitting?

[3]. Thakoor, Shantanu, et al. "Bootstrapped representation learning on graphs." ICLR 2021 workshop.
[4]. Hassani, Kaveh, and Amir Hosein Khasahmadi. "Contrastive multi-view representation learning on graphs." ICML 2020.

---

### Official Review · Reviewer_gZfD · 2025-10-31

**Soundness:** 3
**Presentation:** 3
**Contribution:** 3
**Rating:** 4
**Confidence:** 4

**Summary:**

This paper addresses an important issue in contrastive learning methods, namely the problem of contrastive overfitting. The authors link this problem to the structure-agnostic nature of existing approaches, which discard essential structural information. Empirical results, obtained by integrating a plug-and-play module into existing graph contrastive learning (GCL) models, demonstrate notable performance improvements.

**Strengths:**

- The paper identifies an important problem of contrastive overfitting in graph contrastive learning and links it to the structure-agnostic nature of existing methods, which is a notable contribution.
- It presents extensive experimental analyses, which convincingly demonstrate the novelty and effectiveness of the proposed approach.
- The method is evaluated on various datasets and achieves strong performance on most of them, indicating robustness and general applicability.

**Weaknesses:**

- The positioning of the paper could be further strengthened by incorporating comparisons with recently published related works (e.g., [1], [2], [3]), which would better contextualize the proposed approach within the current research landscape.
- The ablation study could be made more focused and informative. For instance, an ablation examining the individual contribution of each centrality measure would provide clearer insights into the effectiveness of different design choices.
- Evaluating the proposed method against more recent 2025 benchmark baselines would further substantiate its effectiveness and contemporaneous relevance.


[1] Equivalence is All: A Unified View for Self-supervised Graph Learning, ICML 2025
[2] Balancing Graph Embedding Smoothness in Self-supervised Learning via Information-Theoretic Decomposition, WWW 2025
[3] Exploitation of a Latent Mechanism in Graph Contrastive Learning: Representation Scattering, Neurips 2024

**Questions:**

1. It would be helpful if the authors could clarify the derivation of Equation (6), particularly regarding the appearance of the negative sign when transitioning to the third line.
2. The necessity and role of Section 4.2 could be elaborated to understand its contribution to the overall framework better.
3. Is the proposed method effective on more heterophily datasets?
4. In observation Figure 3, isn't it more reasonable to compare the similarity between the target node and its neighbors, rather than comparing with a proxy metric defined in line 195?

Overall, I find the paper interesting and would be open to increasing my score if these points are adequately addressed in the revision.

---

### Official Review · Reviewer_gbD2 · 2025-10-31

**Soundness:** 2
**Presentation:** 3
**Contribution:** 2
**Rating:** 2
**Confidence:** 4

**Summary:**

This paper studies graph contrastive learning (GCL). The authors argue that there exists an overfitting issue and consider it caused by the losing of the structural encoding capabilities of the GNNs with InfoNCE objective. Then, they propose a remedy called Structure-Preserving Regularization (SPR) that preserves node--neighborhood mutual information via a JSD-based objective. They also incorporate a post-hoc augmentation strategy to further improve the performance. The effectiveness is empirically verified through experiments.

**Strengths:**

1. The problem is well-motivated, and is general and critical to the community of graph contrastive learning.

2. The authors present multiple useful empirical observations to explain their hypothesis.

3. Empirically, they apply the proposed method and verify the effectiveness across multiple GCL methods and datasets.

**Weaknesses:**

1. The overfitting issue is too broad and is not limited to GCL (even supervised learning can have the overfitting issue). There are multiple reasons such as the negative sampling as mentioned in this work, spurious correlations[1], and miscalibration[2], etc.

2. The loss of structural preservation can be the consequence of any GNNs that do not perform well, and is not necessarily the reason that caused the observed phenomenon.

3. No theoretical explanation to imply the causal relations between the hypothesis proposed by the authors.

4. The technical novelty is also limited. The proposed methods are simple adaptations of the existing methods.

5. Empirically, the results of the baselines largely differ from those reported in the original paper. It's also unclear how the hyperparameteres are tuned.

6. The experiments do not verify that the proposed method can mitigate the overfitting issue as introduced in the first sections, while just focused on the final performance.


[1] Disentangling Invariant Subgraph via Variance Contrastive Estimation under Distribution Shifts, ICML'25

[2] Calibrating and Improving Graph Contrastive Learning, TMLR'23.

**Questions:**

Please find the details in the weakness section.

---

### Official Review · Reviewer_2aaD · 2025-11-01

**Soundness:** 3
**Presentation:** 3
**Contribution:** 2
**Rating:** 4
**Confidence:** 3

**Summary:**

The paper identifies "contrastive overfitting" in Graph Contrastive Learning (GCL) , where excessively optimizing the contrastive loss leads to worse performance on downstream tasks. The authors attribute this to the structure-agnostic nature of the contrastive objective, which forces the encoder to discard essential structural information.
The authors propose a "plug-and-play" Structure-Preserving Regularization (SPR) framework. SPR introduces a regularization loss to preserve both local and global structural information (like node centrality) and uses a post-hoc message-passing step to re-inject structure into the final embeddings.

**Strengths:**

1. Clearly identifies contrastive overfitting as a novel and significant issue where optimizing GCL loss degrades downstream performance.
2. The motivation is well-explained by multiple empirical observations.
3. The proposed method can be directly used in any model's loss.

**Weaknesses:**

1. The global structure preservation component of SPR requires pre-computing four node centrality measures, including betweenness centrality. Betweenness centrality is very expensive to compute, making this approach potentially unscalable to very large graphs. Also, in the experiments, the graphs used are small graphs instead of large-scale graphs.
2. The authors use four specific centrality measures but provide no justification or ablation for this choice. How critical is each measure? Could comparable performance be achieved with just degree and PageRank, which are much cheaper to compute than betweenness centrality?
3. The paper makes a key point that DGI does not suffer from overfitting because its local-global objective naturally preserves structure. However, DGI is then not included as a baseline in the main experiments. It would be valuable to see if SPR can be applied to DGI and if it offers any further improvement.

**Questions:**

1. The performance on large-scale graphs, including computation efficiency
2. Centrality measures ablation/comparison
3. How is its performance when combined with DGI?

---

### Note · Authors · 2025-11-27

I have read and agree with the venue's withdrawal policy on behalf of myself and my co-authors.